# Silicon and glass very large scale microfluidic droplet integration for terascale generation of polymer microparticles

Sagar Yadavali [1], Heon-Ho Jeong[2,4], Daeyeon Lee[2] & David Issadore[1,2,3]

Microfluidic chips can generate emulsions, which can be used to synthesize polymer microparticles that have superior pharmacological performance compared to particles prepared by conventional techniques. However, low production rates of microfluidics remains a challenge to successfully translate laboratory discoveries to commercial manufacturing. We present a silicon and glass device that incorporates an array of 10,260 (285 × 36) microfluidic droplet generators that uses only a single set of inlets and outlets, increasing throughput by >10,000× compared to microfluidics with a single generator. Our design breaks the tradeoff between the number of generators and the maximum throughput of individual generators by incorporating high aspect ratio flow resistors. We test these design strategies by generating hexadecane microdroplets at >1 trillion droplets per h with a coefficient of variation CV <3%. To demonstrate the synthesis of biocompatible microparticles, we generated 8–16 μm polycaprolactone particles with a CV <5% at a rate of 277 g h$^{-1}$.

[1] Department of Bioengineering, University of Pennsylvania, Philadelphia, PA 19104, USA. [2] Department of Chemical and Biomolecular Engineering, University of Pennsylvania, Philadelphia, PA 19104, USA. [3] Department of Electrical and Systems Engineering, University of Pennsylvania, Philadelphia, PA 19104, USA. [4] Present address: Department of Chemical and Biomolecular Engineering, Chonnam National University, JeonnamYeosu 59626, Republic of Korea. Correspondence and requests for materials should be addressed to S.Y. (email: yadavali@seas.upenn.edu) or to D.I. (email: issadore@seas.upenn.edu)

In 1959, Richard Feynman famously proposed the use of micrometer- and nanometer-scale particles for medicine, as well as the creation of enormous numbers of microfabricated "factories" to generate large quantities of these engineered materials[1]. In the last two decades, significant progress has been made toward accomplishing this vision. In particular, microfluidics has been used to enable precise control of multiphasic flows to generate micrometer- and nanometer-scale materials with control and uniformity not possible using conventional techniques[2–6]. These micro- and nano-engineered materials have generated particular enthusiasm in the pharmaceutical industry, as well as the food and cosmetics industries, where they have created new opportunities to generate novel drug formulations that offer unprecedented spatial and temporal control of drug delivery within the body[7–9]. In comparison to conventional particle formation techniques, such as spray drying or ball milling[10], microfluidic-generated formulations have demonstrated increased particle monodispersity, more uniform composition of drug within the particles, increased drug yield, longer lasting formulations that are still injectable, and reduced burst release of drug[2,6,11].

The low production rate of microfluidic devices for the generation of microparticles ($< 10\ \mathrm{mL\ h^{-1}}$ for the dispersed phase, $< 100\ \mathrm{mg\ h^{-1}}$ of particles) has remained a key challenge to successfully translate the many promising laboratory-scale results of microfluidics to commercial-scale production of microfluidic-generated materials. In previous work, architectures have been developed that make it possible to operate many microfluidic droplet generators in parallel[11–22]. While great progress has been made in these approaches, current chips with parallelized devices are limited to production rates $\phi_{\max} \lesssim 1\ \mathrm{L\ h^{-1}}$, have droplet homogeneities set by three-dimensional (3D) soft-lithography fabrication[15], are limited to low temperature and pressure operation, can only be used with the solvents compatible with the device's polymer construction, or are unable to be adapted to produce higher-order emulsions and particles that require multi-step processing[23–25].

To address these challenges, we present the all silicon and glass very large scale droplet integration (VLSDI), in which we incorporate an array of 10,260 ($285 \times 36$) microfluidic droplet generators onto a 3D-etched single silicon wafer that is operated using only a single set of inlets and outlets. The monolithic construction from a single silicon wafer obviates the alignment and bonding challenges of prior multilayer approaches and allows high pressure use. To demonstrate the power of this approach, we generate polycaprolactone (PCL) solid microparticles, a biodegradable material approved by the United States Food and Drug Administration (US FDA), with a coefficient of variation CV <5%, and an emulsion production rate that results in $277\ \mathrm{g\ h^{-1}}$ particle production ($2.09\ \mathrm{L\ h^{-1}}$ dispersed phase, 328 billion particles per h). Key to achieving this throughput lies in a design strategy that breaks the tradeoff between the number of integrated droplet generators $N_{\max}$ and the maximum throughput of each droplet generator $\phi^{i}_{\max}$. Our design includes a high aspect ratio flow resistor into each droplet generator, which decouples the design of the individual droplet generator from the high fluidic resistance requirement necessary for parallelization. Moreover, because of the VLSDI's all silicon and glass construction, it can operate at high pressure ($P_{\max} > 1000$ PSI (pounds per square inch)) and high temperature ($T_{\max} > 500\ ^\circ\mathrm{C}$), use solvents prevalent in the pharmaceutical industry but that are incompatible with polymer devices, and achieve a uniformity not possible using soft-lithography based devices (CV <3%). Because of the VLSDI's 3D fabrication strategy, it can be scaled to the 10,000 droplet generators that we demonstrate in this study and beyond using conventional semiconductor fabrication, in contrast to prior approaches that have used either two-dimensional microfluidic or two-dimensional microfluidics attached to macroscopically defined manifolds. Because our device allows arbitrary microfluidic droplet generators to be parallelized, it can produce higher-order emulsions and particles that require multi-step processing.

## Results

**Very large scale droplet integration fabrication.** We fabricate the VLSDI using a single microfabricated 500 μm thick 4″ Si wafer encapsulated in glass, resulting in a robust, monolithic construction.(Supplementary Fig. 2 for step by step fabrication) (Fig. 1a, b) This design enables the production of highly monodispersed PCL solid microparticles (CV <5%), >1000× faster than previously reported parallelized microfluidic approaches (Fig. 1c, d).

We use four steps of lithography and deep reactive ion etch (DRIE) to define the droplet generators ($h_{\mu\mathrm{F}} = 22.5\ \mu\mathrm{m}$), underpasses ($h_{\mathrm{UP}} = 26\ \mu\mathrm{m}$), vias ($h_{\mathrm{V}} = 130\ \mu\mathrm{m}$), and delivery channels ($h_{\mathrm{D}} = 360\ \mu\mathrm{m}$) (Fig. 1e). Underpasses are channels that are required to allow fluid to pass underneath the arterial lines that deliver fluid to each of the rows (Supplementary Fig. 1b). The microfabricated Si wafer is anodically bonded to two 4″ Borofloat33 glass wafers on its top and its bottom. Anodic bonding results in microfluidic channels that can operate at a maximum pressure of >1000 PSI, enabling high-throughput operation even on highly viscous samples. To make fluid connections to the VLSDI, we drill 1.5 mm holes in the top glass plate before anodic bonding using an excimer laser micromachining tool (IPG Photonics IX-255). We use steel compression fittings to avoid the creation of debris in the device that can occur using pressure fit tubing (Supplementary Fig. 4). The fittings are bonded to our device using chemically resistant epoxy (Mater Bond Epoxy EP41S-5) and connected to polytetrafluoroethylene (PTFE) tubing (1/8″ OD (outter diameter), 1/16″ ID (inner diameter)). The dispersed and continuous phases are delivered to the device from N$_2$ gas pressured steel pressure vessels (Alloy Products) through the PTFE tubings. All of the parts and tubing are chemically resistant, allowing our platform to generate particles using a wide range of temperatures, pressures, and solvents (Supplementary Figs 3–5).

**Design principles.** There are three main design goals for our parallelized device: to ensure uniform flow across each of the $N$ microfluidic droplet generators, such that each generator supplies the same shear stresses to produce droplets with the same diameter. To minimize the footprint of each individual microfluidic droplet generator, such that the greatest number $N$ can be incorporated onto a given wafer. To maximize the production rate (droplets per s) of each individual droplet generator, while keeping the microfluidic device in a low flow velocity regime where each device produces uniform droplets (i.e., a dripping regime not a jetting regime)[26]. These design goals and the fluid physics of microfluidic droplet generators provide tradeoff relationships and constraints that guide the VLSDI's design.

The droplet generators on our device use a flow-focusing geometry (Fig. 1b) and are organized in a ladder architecture[13–16] (Fig. 1f) (Supplementary Fig. 1). In the ladder design, the individual generators are connected in a line along a single set of liquid distribution channels. A two-dimensional array of microfluidic droplet generators is created by connecting multiple rows of droplet generators, also in a ladder geometry, using a single set of arterial lines. In previous work, it has been shown that liquids can be uniformly distributed over $N$ flow-focusing generators (FFGs) connected using a single set of distribution channels, such that the flow resistance in the distribution channel between droplet generators $R_{\mathrm{D}}$ is small relative to that of each individual

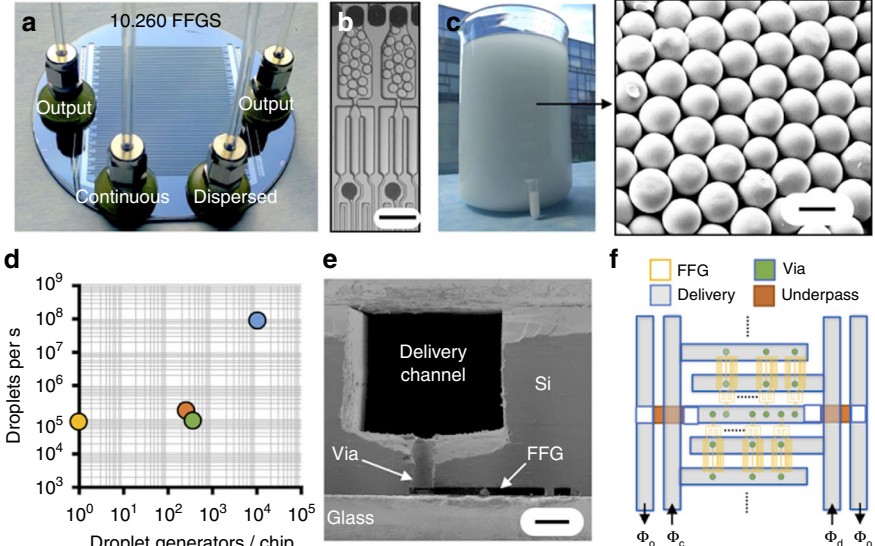

**Fig. 1** All silicon and glass VLSDI for high-throughput microparticle production. **a** A photograph of the VLSDI chip, which consists of 10,260 (285 × 36) flow-focusing droplet generators (FFGs). **b** An optical micrograph of two individual droplet generators producing droplets. Scale bar: 140 µm. **c** Organic phase in water emulsion (3.5 L) generated from VLSDI in 30 min on a chip operated at a dispersed flow rate $\Phi_d = 2.09$ L h$^{-1}$ (10 wt% polycaprolactone in dichloromethane) and a continuous flow rate $\Phi_c = 5$ L h$^{-1}$(2 wt% polyvinylalcohol in water). The inset shows an SEM (scanning electron microscope) image of polycaprolactone microparticles obtained after evaporation of dichloromethane. Scale bar: 8 µm. The 2 mL tube represents the emulsion collected from a conventional microfluidic droplet generator in 1 h. **d** Moore's Law of polymer microparticle generation. Yellow[11], green[19], orange[12], blue: this work. **e** An SEM micrograph of the cross section of the VLSDI chip. Scale bar: 90 µm. **f** The ladder design architecture that our chip uses to incorporate 10,260 parallel droplet generators. $\Phi_o$ emulsion output, $\Phi_c$,$\Phi_d$ supply channels for continuous and dispersed phase

microfluidic droplet generator $R_{Dev}$, resulting in a design rule[16]:

$$2N(R_D/R_{Dev}) < 0.01. \qquad (1)$$

In previous work, this design rule has been satisfied for large numbers of microfluidic droplet generators by taking advantage of the hydrodynamic resistance of a microfluidic channel's $R \propto 1/h^3$ dependence on the height $h$ of the microfluidic channel, using delivery channels with a height $h_D > 100$ µm and droplet generators with a height $h_{Dev}$ ~10 µm[13–16]. Using these design principles for parallelizing microfluidic droplet generators, there is a tradeoff between the number of droplet generators that can be incorporated onto a given chip area $N$ and the maximum throughput $\phi$ of each individual droplet generator. The origin of this tradeoff comes from two competing goals: *a* to keep the fluid velocity low at high volumetric flow rates, thus decreasing the device's fluidic resistance $R_{Dev}$, such that each device produces uniform droplets (i.e., a dripping regime not a jetting regime)[26] and *b* to increase the fluidic resistance $R_{Dev}$ of each droplet generator to satisfy Eq. 1 for the largest possible number of droplet generators $N$.

The transition from dripping to jetting is a well-studied phenomenon, and depends on the capillary number of the continuous phase of each individual flow-focusing droplet generator $Ca_o = \mu v/\sigma$ (where $\mu$ is the flow viscosity, $v$ is the velocity, and $\sigma$ is the interfacial tension), representing the relative importance of surface tension to viscous forces, as well as the Weber number $We_D$ of the dispersed phase, which represents the relative magnitudes of inertial and surface tension forces. When Ca and We $< O(10^0)$, surface tension force dominates the droplet break-up and uniform droplets are formed via the dripping mechanism[26]. For the application of high-throughput particle production, the dripping to jetting transition defines a maximum throughput $\phi^i_{max}$ for the individual droplet generators.

To overcome the tradeoff between increasing the number of droplet generators $N$ and the maximum flow rate at which each device can be operated $\phi^i_{max}$, we incorporated flow resistors for both the dispersed and continuous phase upstream of the droplet generators (Fig. 2a). Each of these flow resistors has a width less than their height $w < h$ ($w = 10$ µm, $h = 22.5$ µm), such that the dependence of resistance on the channel dimensions become $R_R \propto 1/hw^3$. By adding these flow resistors, the resistance of the individual droplet generators $R_{Dev}$ decreases as a function of $h$ at a rate less than that of a traditional parallelization design $R_{Dev} \propto 1/wh^3$[13,14,16]. Thus, the minimum resistance $R^{min}_{dev}$ necessary to incorporate 10,260 FFGs to satisfy Eq. 1 can be achieved at a greater height $h$ (Fig. 2e) and thus enable the droplet generators to operate at a higher volumetric flow rate before transitioning from the dripping to jetting regime. This work, wherein we have decoupled the design of individual flow-focusing droplet generators in massively parallelized chips, builds on earlier reports where flow resistors were used to control the flow rate and decouple droplet generators in chips that contain several droplet generators[27,28].

**Validation of design principles**. To validate the design principles described above, we analyzed the performance of three VLSDI devices whose channel dimensions satisfy the design rule for 10,260 FFGs (Eq. 1) (Fig. 2a). For each design, the footprint of the individual droplet generator is identical (175 µm × 1475 µm) and 10,260 FFGs are incorporated using the same ladder geometry. Device I: we built this device without the benefit of our decoupling strategy, using upstream flow resistors, with height $h = 10$ µm and the width $w = 20$ µm immediately downstream of the droplet generator and $w = 10$ µm upstream. Device II: here, we incorporated flow resistors and increased the dimensions of the droplet generators to $h = 22.5$ µm and $w = 80$ µm immediately downstream of the droplet generator and $w = 40$ µm upstream in the dispersed and continuous phases, and Device III: we increased

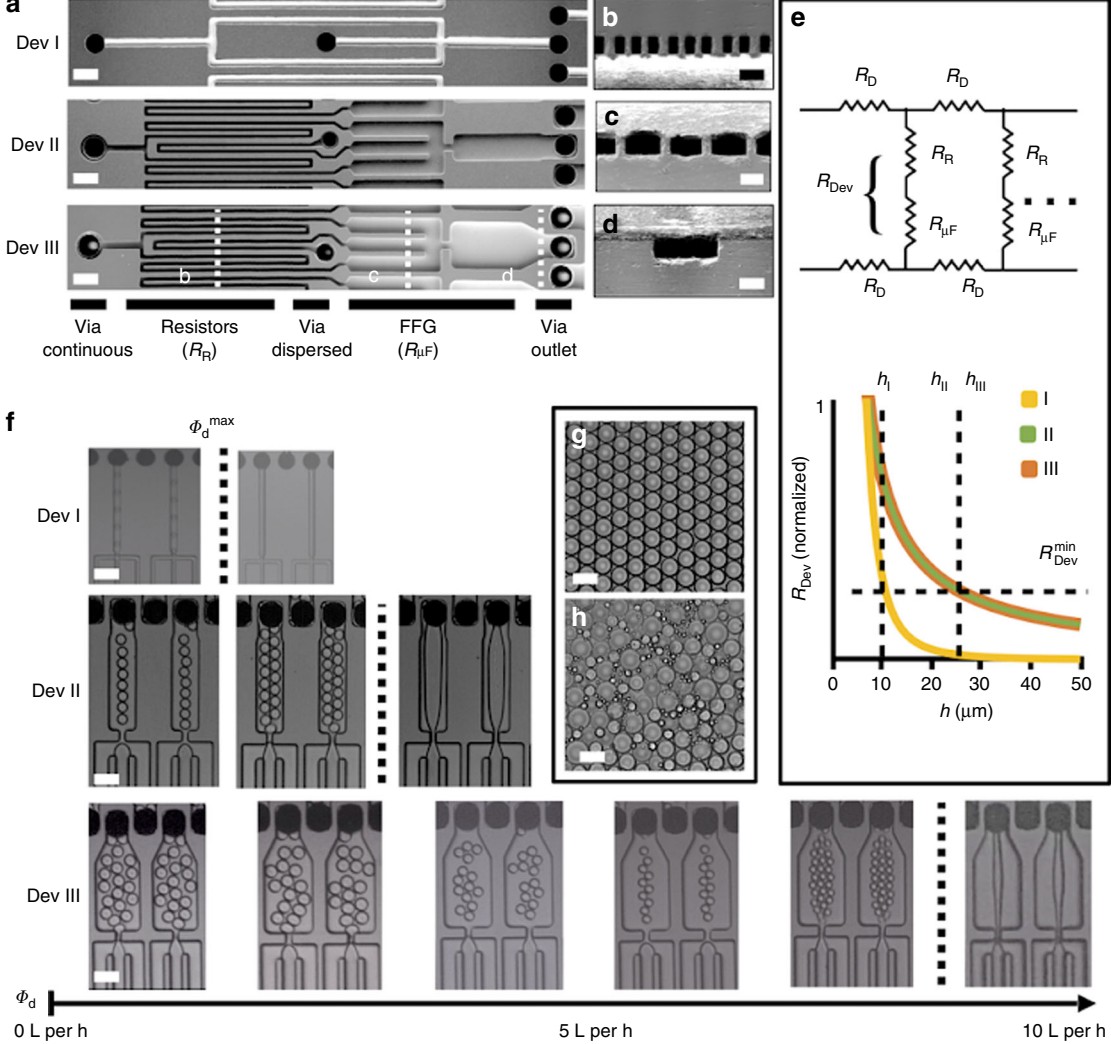

**Fig. 2** Individual droplet generator design for high-throughput emulsion generation. **a** SEM images of the three flow-focusing droplet generator designs used in our study. Each FFG design occupies the same footprint (175 μm × 1475 μm), and for each design 10,260 were incorporated onto a VLSDI chip using the ladder architecture. Scale bars of all SEM images: 65 μm. **b–d** shows cross-sections of Dev III. Scale bar: 30 μm. **e** A lumped circuit model of our droplet generators. The graph shows the resistance of an individual droplet device $R_{Dev}$ vs. the channel height $h$. The inclusion of high aspect ratio resistors in Dev II and Dev III, leads to $R_{Dev}$ decreasing at a rate less than $R_{Dev} \propto 1/h^3$ of the traditional design Dev I. Thus, the minimum resistance $R_{dev}^{min}$ necessary to incorporate 10,260 FFGs can be achieved at a greater height $h$ for Dev II and Dev III that enable operation at a higher flow rate before transitioning from the dripping to jetting regime. **f** Optical micrographs of droplet generators producing hexadecane droplets in water at various dispersed phase flow rates $\phi_d$ for Dev I, II, and III. The maximum flow rate $\phi_d^{max}$ possible, while keeping the devices in the dripping regime, using Dev III ($\phi_{max} = 7.3 \, \text{L h}^{-1}$) is greater than Dev II ($\phi_{max} = 3.5 \, \text{L h}^{-1}$), which is greater than Dev. I ($\phi_{max} = 0.35 \, \text{L h}^{-1}$). Scale bars of all images: 65 μm. **g** An example of droplets formed when in the dripping regime (scale bar: 30 μm) and in the jetting regime (**h**) (scale bar: 40 μm)

the dimensions of the droplet generator further, such that $h = 22.5$ μm and $w = 140$ μm immediately downstream of the droplet generator and $w = 40$ μm upstream in the dispersed and continuous phases. The flow resistors, with width $w = 10$ μm and height $h = 22.5$ μm, are incorporated in the same layer as the microfluidic droplet generators. The channel length between the flow resistor and the droplet generator is defined by the low Reynold's number entrance length (length $= 0.06 \times \text{Re}$)[29].

We first evaluated these devices by generating hexadecane droplets in water (2 wt% Tween 80). We confirmed that at all flow rates droplets are generated in every one of the 10,260 droplet generators (Supplementary Movie 1). As expected, we found that Device I transitioned from making uniform droplets to polydisperse droplets at a low dispersed flow rate of $\phi_d^{max} = 0.35 \, \text{L h}^{-1}$ (Fig. 2f), Device II with its increased channel

dimensions produced uniform droplets at a maximum dispersed flow rate of $\phi_d^{max} = 3.5 \, \text{L h}^{-1}$, and Device III with its even larger downstream channel dimensions produced uniform droplets up to a maximum dispersed flow rate of $\phi_d^{max} = 7.3 \, \text{L h}^{-1}$. For each of the three devices, at flow rates where the device was in the dripping regime, the droplets were highly monodispersed (CV <5%) (Fig. 2g). And, at flow rates where the droplet generator were in the jetting regime, the droplets became highly polydisperse (CV »5%) (Fig. 2h). All three devices transitioned from dripping to jetting at approximately the same Capillary number Ca immediately downstream of the droplet generator orifice (Ca≅0.08).

We tested the mass production of oil-in-water emulsion by using pressure-driven flow (Supplementary Figs. 4 and 5 for experimental setup) and the size of emulsion droplets could be

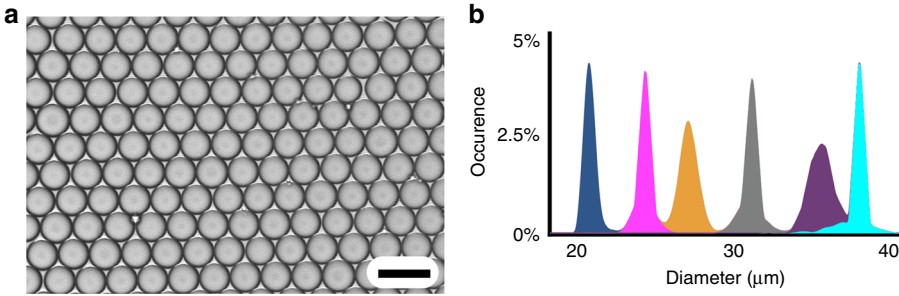

**Fig. 3** High-throughput production of highly monodisperse hexadecane droplets from Dev III-VLSDI. **a** An optical image of hexadecane droplets with $d =$ 22.5 μm (CV = 3.30%) at $\Phi_d = 7.30$ L h$^{-1}$. **b** The droplet diameter could be controlled by varying the flow rate of the dispersed and continuous phase fluids. Droplets were produced over a range of $d = 22.5$–37.5 μm, all with CV ~3%. The dispersed phase $\Phi_d$ is hexadecane and the continuous phase $\Phi_c$ is water with 2 wt% Tween 80. Blue: $d = 22.5$ μm, CV = 3.30%, $\Phi_d = 7.30$ L h$^{-1}$, $\Phi_c = 9.20$ L h$^{-1}$; magenta: $d = 25.5$ μm, CV = 2.30%, $\Phi_d = 0.40$ L h$^{-1}$, $\Phi_c = 5.60$ L h$^{-1}$; orange: $d = 28.5$ μm, CV = 2.92%, $\Phi_d = 4.00$ L h$^{-1}$, $\Phi_c = 6.25$ L h$^{-1}$; gray: $d = 31.6$ μm, CV = 2.62%, $\Phi_d = 1.00$ L h$^{-1}$, $\Phi_c = 2.80$ L h$^{-1}$; purple: $d = 35.5$ μm, CV = 1.93%, $\Phi_d = 1.08$ L h$^{-1}$, $\Phi_c = 1.50$ L h$^{-1}$; turquoise: $d = 37.5$ μm, CV = 1.75%, $\Phi_d = 1.20$ L h$^{-1}$, $\Phi_c = 1.40$ L h$^{-1}$

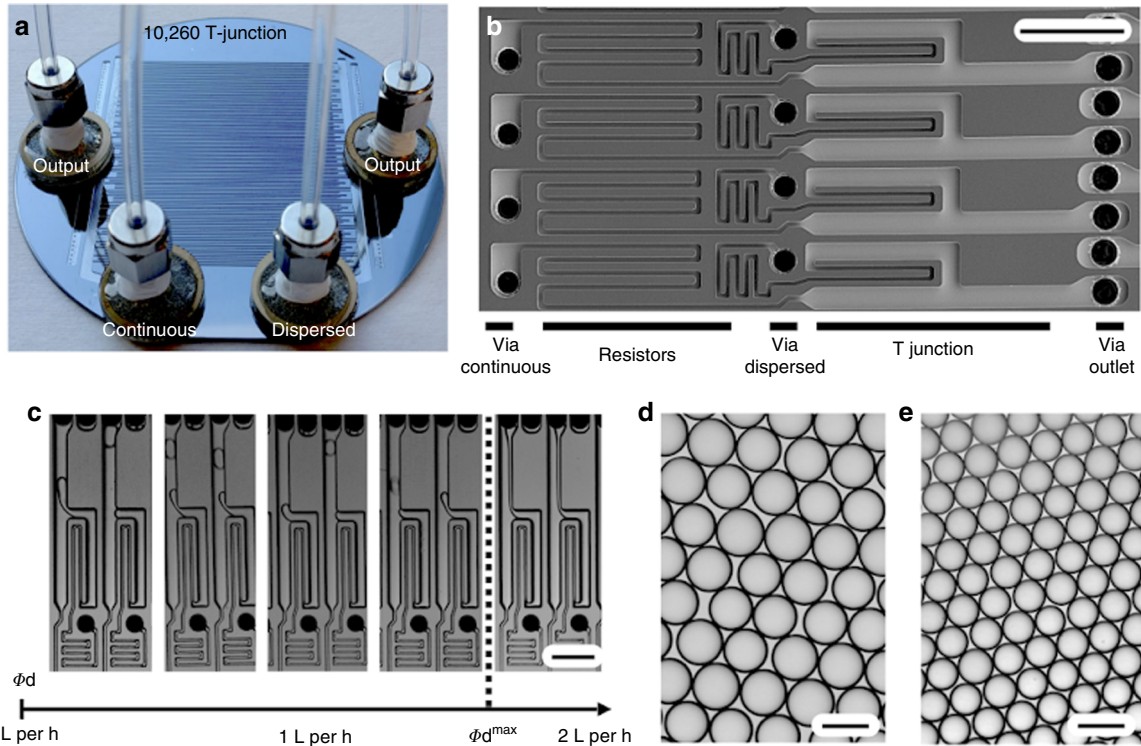

**Fig. 4** A VLSDI chip that contains 10,260 (285 × 36) T-junction devices. **a** A photograph of the T-junction VLSDI chip. **b** An electron micrograph of four individual droplet generators. Scale bar: 250 μm. **c** Optical micrographs of T-junction droplet generators producing hexadecane droplets in water at various dispersed phase flow rates $\phi_d$. The maximum flow rate $\phi_{max}$ possible. Scale bars of all images: 100 μm. A micrograph of hexadecane droplets in water produced by the T-junction, **d** which have an average diameter $d = 46.2$ μm and a coefficient of variation CV = 3.73% and **e** which have an average diameter $d = 33.2$ μm and a coefficient of variation CV = 3.27 %. Scale bar: 50 μm

changed by varying the ratio of the flow rates of the dispersed and continuous phases (Supplementary Movie 2). For example, by changing the dispersed oil phase flow rate over the range of $\phi_d =$ 0.5 L h$^{-1}$ (10 PSI) to 7.3 L h$^{-1}$ (59 PSI) and the continuous aqueous phase over the range of $\phi_c = 0.8$ L h$^{-1}$ (12 PSI) to 9.2 L h$^{-1}$ (62 PSI), the average droplet size could be controlled over a range of $d = 22.5$–37.5 μm (Fig. 3a, Supplementary Fig. 6). The generated droplets were highly monodisperse at all flow rates, with the coefficient of variation CV <3% (Fig. 3b) at a throughput >1 trillion droplets per h (Supplementary Movie 3). Because our device is fabricated using anodically bonded silicon and glass, it can operate at

extremely high pressures (>1000 PSI), making it very well suited for high-throughput processing of highly viscous fluids. We demonstrated the high-throughput production of mineral oil droplets (30 cP), which is an order of magnitude more viscous than the hexadecane (3.0 cP) and dichloromethane (DCM) (0.41 cP). We achieved a dispersed phase throughput of $\phi_D = 2.2$ L h$^{-1}$ ($d = 24.3$ μm droplets, CV <3%) (Supplementary Fig. 7 and Supplementary Movie 4). The size of the droplets that could be produced by each of the chips were comparable to that of the orifice dimensions, as has been shown in previous work[30]. Using our Device III chip, which had orifice dimensions of height $h = 22.5$ μm and width $w = 20$ μm,

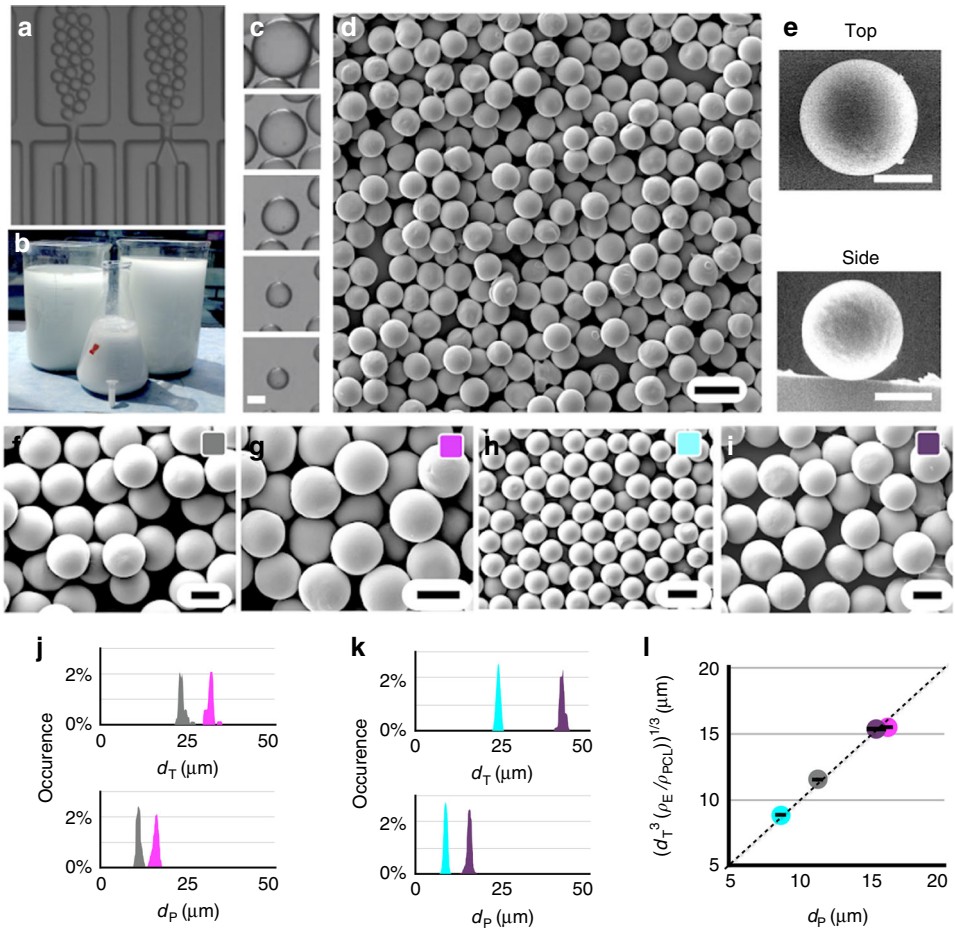

**Fig. 5** High-throughput production of polycaprolactone microparticles. **a** Optical image of droplet generators producing monodisperse emulsion templates. The dispersed phase $\Phi_d$ is 133 g L$^{-1}$ of polycaprolactone (PCL) in dichloromethane (DCM), and continuous phase $\Phi_c$ is 2 wt% of polyvinyl alcohol (PVA) in water. **b** Emulsion of 8.76 L could be generated in 75 min. **c**. Evaporation of dichloromethane from emulsion templates as a function of time. Scale bar: 10 μm. **d** An SEM micrograph of PCL microparticles, produced at 277 g h$^{-1}$. Scale bar: 20 μm. **e** Top and side view of a single PCL microparticle. Scale bar: 10 μm. **f–i** SEM micrographs of PCL microparticles generated with particle diameter $d_P$ using a template diameter $d_T$. Scale bar: 10 μm. Using 133 g L$^{-1}$ PCL, we generated particles: **f** $d_P = 11.2$ μm (CV = 4.4%), $d_T = 23.2$ μm (CV = 2.5%) at 277 g h$^{-1}$ and **g** $d_P = 16.1$ μm (CV = 4.4%), $d_T = 35.9$ μm (CV = 1.4%) at 109 g h$^{-1}$. Using 53.2 g L$^{-1}$ PCL, we generated particles: **h** $d_P = 8.4$ μm (CV = 4.0%), $d_T = 24.5$ μm (CV = 2.4%) at 76.8 g h$^{-1}$ and (**i**) $d_P = 15.2$ μm (CV = 3.6%), $d_T = 42.7$ μm (CV = 2.0%) at 38.5 g h$^{-1}$. Histograms of template diameters and particle diameters produced using 133 g L$^{-1}$ PCL (**j**) and 53.2 g L$^{-1}$ PCL (**k**). **l** The measured particle diameters $d_T$ agreed well ($R^2 = 0.99$) with the particle diameter predicted based on the template diameter $d_T$, $\rho_E$ is the wt/vol concentration of PCL in our emulsions, and $\rho_{PCL} = 1.143$ gmL$^{-1}$ is the density of solid PCL. The gray line represents ideal agreement between the particle size $d_P$ and the prediction

we were able to produce droplets with diameters $d = 22.5$–$37.5$ μm. With the Device I chip that had orifice dimensions of height $h = 10$ μm and width $w = 10$ μm, we were able to produce droplets with diameters $d = 14.5$–$18.4$ μm.

**VLSDI integration of T-junctions**. To demonstrate the modularity of the VLSDI architecture, we designed, fabricated, and tested a version of the chip that incorporates 10,260 (285 × 36) T-junction devices (Fig. 4a). We designed flow resistors (width $w = 10$ μm and height $h = 18$ μm) upstream of the T-junctions, following the same design rules used on our FFG device (Fig. 4b). We first evaluated these devices by generating hexadecane droplets in water (2 wt% Tween 80) and confirmed that at all flow rates droplets are generated in every one of the 10,260 droplet generators (Supplementary Movie 5). We found that the T-junction device could generate droplets at a maximum dispersed flow rate of $\phi_d^{max} = 1.5$ L h$^{-1}$ (Fig. 4c). We mass produced oil-in-water emulsion using this device by using pressure-driven flow and the size of emulsion droplets could be changed by varying the ratio of the flow rates of the

dispersed and continuous phases. Droplets were generated with sizes that range from 46.2 μm ($\phi_d = 0.5$ L h$^{-1}$, $\phi_c = 4.5$ L h$^{-1}$) (Fig. 4d) to 33.2 μm ($\phi_d = 1.5$ L h$^{-1}$, $\phi_c = 12.8$ L h$^{-1}$) (Fig. 4e). The generated droplets were highly monodisperse with the coefficient of variation CV <4% at a throughput of 80 billion droplets per h.

**Large-scale manufacturing of polymer microparticles**. To demonstrate the power of this approach for the industrial scale manufacturing of uniform solid particles, appropriate to use as an injectable drug delivery system[5,6], we generated micrometer scale PCL microparticles, a biodegradable material approved by the US FDA[31]. We achieved a production rate of 277 g h$^{-1}$ (328 billion particles per h) for particles with diameters ranging from 8–15 μm with CV <5% (Supplementary Movie 6, Supplementary Movie 7). Emulsion templates for the solid particles were fabricated on our chip using a dispersed phase of DCM with either 4 wt% ($\rho_E = 53.2$ g L$^{-1}$) or 10 wt% ($\rho_E = 133$ g L$^{-1}$) of PCL. The continuous phase was deionized water with 2 wt/vol% of polyvinyl alcohol (PVA) (Fig. 5a). These two phases were driven through the FFG-based VLSDI chip

using pressure-driven flow, collected, and then further processed using roto-evaporation and lyophilization prior to being analyzed. Uniform DCM droplets were generated with sizes ranging from 23 to 42 μm with CV <3% at a production rate of $\phi_d \cong 2.09$ L h$^{-1}$ (Fig. 5b). After the DCM was extracted (Fig. 5c), spherical, highly monodispersed solid PCL polymer particles remain (Fig. 5d, e).

To test our chip's ability to produce highly monodispersed particles with a determined particle diameter, we generated four different particle formulations. Using a dispersed phase of DCM with 10 wt% PCL ($\rho_E = 133$ g L$^{-1}$), we generated two pools of particles, one with a diameter of $d_p = 11.2$ μm (CV = 4.4%) (Fig. 5f) and one with a diameter $d_p = 16.1$ μm (CV = 4.4%) (Fig. 5g), beginning with droplet templates with diameters of $d_T = 23.2$ μm (CV = 2.5%) and $d_T = 35.9$ μm (CV = 1.4%), respectively. Using a dispersed phase of DCM with 4 wt% PCL ($\rho_E = 53.2$ g L$^{-1}$), we generated two populations of particles, one with a diameter of $d_p = 8.4$ μm (CV = 4.0%) (Fig. 5h) and one with a diameter $d_p = 15.2$ μm (CV = 3.6%) (Fig. 5i), beginning with droplet templates with diameters $d_T = 24.5$ μm (CV = 2.4%) and $d_T = 42.7$ μm (CV = 2.0%), respectively. The slight increase in CV from droplets to particles came primarily from rare particles that were deformed during post-processing off chip, and we postulate that CV can be further improved by translating our process to a continuous flow liquid-liquid extraction system[32]. We have produced as much as two gallons of DCM-PCL in water emulsion droplets at a flow rate of $\phi_d = 2.09$ L h$^{-1}$ with a run time of 75 min without a single device failing. These tests were limited by the size of our pressure vessels—1 gallon for dispersed, 3 gallons for continuous phase (Supplementary Fig. 5).

We compared the measured diameter of our microparticles $d_p$ with their expected diameter $d^3_p = d_T{}^3 (\rho_E/\rho_{PCL})$ based on the diameter of the emulsion template $d_T$, the density of solid PCL microparticle $\rho_{PCL} = 1.143$ g/mL, and the weight per volume concentration of PCL in our emulsions $\rho_E$. We considered four separate formulations, generated using a dispersed phase with both $\rho_E = 133$ g L$^{-1}$ (Fig. 5j) and $\rho_E = 53.2$ g L$^{-1}$ (Fig. 5k). Excellent agreement was found between the measured particle diameters and the predicted diameters ($R^2 = 0.99$), suggesting the particles prepared from the emulsions were non-porous. (Fig. 5l).

## Discussion

We present a new platform to mass produce highly uniform microparticles using a highly parallelized microfluidic device. By developing a new 3D architecture, implemented entirely in a monolithically fabricated silicon wafer with glass encapsulation, we can generate polymer solid microparticles at a rate >1000× faster than existing parallelized devices[12,21,34]. Moreover, if commercialized and implemented using a 12-inch wafer[34], a production rate of 10 trillion droplets per h is feasible. In this paper, we systematically studied the effect of flow rate on droplet uniformity and demonstrated that the addition of high aspect ratio flow resistors, allows microfluidic droplet generators that can operate at high production rates to be successfully parallelized without redesign.

Given the high value of therapeutics and the large mass of drug particles that each VLSDI device could produce, we postulate that these devices could be economically feasible for use in the pharmaceutical industry[21]. We estimate that if these chips were fabricated at-scale (>1000), then each wafer would cost on the order of a hundred dollars. As an example, the current market for anti-retroviral therapy (ART), for the 37 million people living with HIV worldwide, is ~$24 billion[35]. To manufacture the entire world's supply of ART in the form of long-lasting microparticle-based injectables, assuming 200 mg drug administration per day per patient and a 200 g per h production rate and a fraction of active ingredient in each particle of 10–70%, <100 of our chips

continuously running 24 h a day could provide the world's supply. Moreover, due to the low cost and automated use of the VLSDI design, it can be used for point-of-demand pharmaceutical production at locations closer to the patient. An improved solution for automated manufacturing of high-quality pharmaceutical formulations closer to the point of patient care can help address current challenges, such as the shortage of generic injectables[36] associated with high manufacturing and storage costs, which currently limit access to essential therapies for sepsis, cancer, and other life threatening conditions.

## Methods

**VLSDI device fabrication.** The VLSDI was fabricated at The Singh Center at The University of Pennsylvania (Supplementary Fig. 2). The designs for all layers of the VLSDI chips were designed in DraftSight (Dassault Systems). Four mask layers were designed, droplet makers (Layer-1), underpass channels (Layer-2), vias (Layer-3), and delivery channels (Layer-4). The designs files were written on chrome-coated soda lime photomasks (AZ1500) using Heidelberg 66 plus mask writer with a 10 mm write head. After exposure, the photomasks were developed in MF 319 for 1 min and then in Chrome etchant for 1 min. Finally, the remaining photoresist on the photomask is removed by plasma oxidation for 10 min in Anatech SCE-106-barrel Asher.

All layers of the VLSDI chip were lithographically patterned and etched in a single 4-inch double-side polished silicon wafer using DRIE (SPTS Rapier Si DRIE). In the first etch step, 16 μm of positive photoresist SPR 220.7 is spin-coated on the front side of the silicon wafer, and exposed with the delivery channels (Layer-4) photomask. After exposure, the wafer is left at room temperature for rehydration for 24 h, then it is developed in MF 319 for 1 min. The wafer is then etched in DRIE for an etch depth of 370 μm. The wafer is then cleaned in Acetone and placed in nanostrip for 30 min and cleaned in DI water and dried in N$_2$ gas. In the second etch step, the wafer is flipped and spin-coated with 12 μm SPR 220.7 positive photoresist. The wafer is then exposed with Via layer (Layer-3) photomask, and left at room temperature for 12 h for rehydration, and then developed in MF-319 for 1 min. The wafer is bonded to another silicon wafer (carrier wafer) using crystal bond adhesive. The wafer is etched in DRIE for through vias in the silicon wafer. The wafer is then placed on a hotplate at 65 °C, and the carrier wafer is removed. The wafer is cleaned in acetone, and then nanostrip, and then in DI water, and then dried in N$_2$ gas. In the third etch step, the wafer is spray-coated with 8 μm S1805 resist. The wafer is then exposed with underpass channel (Layer-2) photomask, and then developed in MF 319. The wafer is again bonded to carrier wafer using crystal bond. Underpass channels are etched in DRIE for a 30 μm deep etch. The wafer is then cleaned in acetone, and nanostrip. In the final etch step, the wafer is spray-coated with 4 μm S1805 resist, and then exposed with the droplet maker photomask (Layer-1), and developed in MF 319 for 1 min. The wafer is placed on a carrier wafer, and then etched for 23 μm deep in DRIE. The carrier wafer is removed by placing it on hotplate at 65 °C. The 3D-etched silicon wafer is then placed in acetone and in nanostrip for 1 h. Borofloat33 glass wafers are laser micromachined with 1 mm holes using IPG Photonics excimer laser. These vias serve as inlets and outlet connections for the VLSDI chip.

The completely etched silicon wafer and plain Borofloat33 glass wafer is cleaned in acetone, Isopropyl alcohol (IPA), and deionzied water (DI) water for 10 min each. The wafers are then dried in a Spin Rinse dryer. The wafers are then placed in nanostrip for half an hour and then placed in piranha solution for 2 h. Then, the wafers are cleaned in water, dried in spin rinse dryer, and then bonded in an EVG 510 wafer bonder by applying 600 V, and a pressure of 600 N for 1 h. Once front side of wafers are bonded, the bonded wafer and laser machined Borofloat33 wafer are kept in Nanostrip for half an hour and then in piranha solution for 2 h. The wafers are then placed in DI water for 12 h to completely remove the the acid solution that may filled in the channels. The wafers are then dried in spin rinse drier and bonded in EVG 510 wafer bonder with pressure of 600 N for 2 h.

Stainless steel compressed tube fittings (1/8 tube OD) form McMaster Carr (52245K609) are bonded to the glass wafer using chemically resistant epoxy from master bond (EP41S-5). The epoxy is left to cure at room temperature for 4 days. PTFE tubes of 1/8 OD were connected to the fittings.

**Experimental setup.** Pressure-driven flow is used to conduct the experiments. Nitrogen pressure tanks were connected to 1-gallon and 3-gallon stainless steel pressure vessels (Alloy products). The 1-gallon vessel is used for dispersed phase and the 3-gallon vessel is used for continuous phase. The VLSDI chip is connected to the pressure vessels using PTFE tubings. The VLSDI chip is housed in a custom-built acrylic box and then mounted on to an xyz stage. Inline filters (McMaster Carr: 9816K72) are used to filter debris in the continuous and dispersed phases. An inline flow meter (McMaster Carr: 5084K23) was used to measure the flow rates for the aqueous phase. To test hexadecane droplets in water, hexadecane of viscosity $\mu = 3$ cps was purchased from Alfa Aesar (Stock number: 43283-LW). Tween 80 surfactant was purchased from Fisher Scientific (catalog number:

AC278630025). For mineral oil droplets in water, mineral oil with viscosity of $\mu = 30$ cps was purchased from Sigma-Aldrich (product number: M3516).

**Polymer microparticle synthesis**. For the generation of polymer microparticles, DCM solvent of viscosity $\mu = 0.43$ cps was purchased from fisher scientific (catalog number: AC406920040), PCL was purchased from Sigma-Aldrich (product number: 440752) and PVA was purchased from Sigma-Aldrich (product number: 363170). The dispersed phase consisted of 10 wt% (133.3 g/L) or 4 wt% (53.2 g/L) PCL in DCM. DCM of 2 L was mixed with the corresponding amount of PCL and mixed on a magnetic stirrer for 1 h. For the continuous phase, 2 wt% of PVA (87% hydrolyzed) was mixed with water at 95 °C for 12 h. Continuous phase of 9 L was used for each experiment. The generated emulsion templates were collected in 4 L glass beakers. To post-process the emulsion templates, 10 mL of emulsion templates were dispersed in 30 mL of water with PVA and placed in Rotovap for 5 min. The precipitated polymer particles were washed thrice in water using centrifuge and then dried in lyophilization unit for 24 h. Scanning electron microscope images were obtained using JEOL 7500F HRSEM and Quanta 600 FEG SEM. An accelerating voltage of 5 kV is used to collect the SEM images.

**Data availability**. The authors declare that the data supporting the findings of this study are available within the paper and its supplementary information files.

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

## Acknowledgements

We thank Noah Clay (Director), Meredith Metzlerm, and all QNF Staff at University of Pennsylvania for their help in device fabrication. We also thank Ravi Yellesarapu, Hari Katepalli, Martin F. Hasse, Jessica Liu, and Harsha Kalluru for helpful discussions. We are grateful to Dr. Andrew Tsourkas lab—Ahmad Amirshaghaghi, Kido Nwe, and Elizabeth Higbee—for their help in post-processing of polymer emulsion templates. We would like to acknowledge support from The National Science Foundation (1554200) and Glaxo Smith Kline, in particular from David Lai, Sonja Sharpe, and the entire GSK microfluidics team. D.L. acknowledge the support from NSF CBET 1604536.

## Author contributions

S.Y. conceived and performed all designs, fabrication, experiments, and characterization in this study, as well as prepared the manuscript and figures. S.Y. and H.-H.J. performed hexadecane in water experiments. D.L. and D.I. conceived and oversaw all aspects of this study, and prepared the manuscript.

## Additional information

**Competing interests:** David Issadore is the founder of, and currently holds shares of, Chip Diagnostics. The remaining authors declare no competing interests.

