## [Peer Review File(PDF 8255 kb) · Nature Communications]

Reviewers' Comments:

Reviewer #1:

Remarks to the Author:

This paper reports a highly parallelized microfluidic device made of silicon and glass that produces micrometer-sized drops with a narrow size distribution. Drops are produced at a throughput that is much higher than what could previously be achieved. The approximately 1000-fold increase in throughput is achieved by introducing flow resistors that make hydrodynamic resistance of the channel less dependent on its height because it mainly depends on the resistor width. The size of the resulting particles can be controlled to some degree by controlling the fluid flow rates.

The main limitation of microfluidic devices that hinders their use in many scientific applications and almost all industrial applications is their low throughput. This paper presents an approach that is scalable and has thus has the potential to overcome this limitation. The paper is well written and nicely illustrated. Therefore, I recommend accepting the paper after the authors addressed the following comments.

The authors show that by changing the resistor width, they can change the maximum throughput of the device. Does the resistor width influence the drop size or drop size distribution? And how does the channel height influence the drop size?

The authors use low viscosity fluids to test the performance of this device. In many cases, microparticles are made from fluids that are more viscous. Can fluids with higher viscosities also be processed and if yes, up to what viscosity? How does the maximum throughput of the device depend on the viscosity of the inner and outer phase?

Reviewer #2:

Remarks to the Author:

This manuscript reports large-scale integration of microfluidic droplet generators. The authors lithographically microfabricated ~10,000 flow-focusing droplet generators in a matrix array on a glass/silicon chip and demonstrated mass-production of highly monodisperse emulsion droplets and particles.

This work is nicely done and solid scientifically. The text is logically written having few confusing points. However, the reviewer does not consider this work represents important advances of significance necessary for publication in Nature communications.

(1)

There have been several reports on similar parallelization of microfluidic droplet generators in matrix arrays from the authors' group (Lab Chip 2013, 13, 4750; Lab Chip 2015, 15, 4387) and a few other groups (Ind. Eng. Chem. Res. 2009, 48, 8881; Lab Chip 2012, 12, 802). From the reviewer's viewpoint, it seems that the design principle described in the present work is almost the same as these previous works. In particular, the authors previously reported parallelization of ~1000 flow-focusing droplet generators in a polymer device by soft lithography. The increase of the number up to ~10,000 based on similar approach with different device materials (glass/silicon) and conventional lithographical processes does not seem to be a significant advance.

(2)

The authors claim that "In contrast to prior parallelization approaches, our design breaks the trade-off...by incorporating high aspect ratio flow resistors.". However, the insertion of flow resistors in front of each of parallel droplet generators is quite normal (e.g., see Biomicrofluidics 2013, 7, 034112; Biomicrofluidics 2015, 9, 034101).

(3)

The use of non-polymer materials like glass and silicon for fabricating parallel microfluidic devices for generating droplets and particles is not new (e.g., Ind. Eng. Chem. Res. 2002, 41, 4043; Lab Chip 2008, 8, 287; Lab Chip 2012, 12, 3426; Lab Chip 2015, 15, 2486; Macromol. Chem. Phys. 2017, 218, 1600472, etc.) . It is not surprising to use non-polymer materials together with conventional microfabrication processes although their cost is higher than those of polymer devices.

Reviewer #3:

Remarks to the Author:

The manuscript describes the performance of a high-throughput microfluidic drop generation system. Microfluidic systems with more than 10,000 flow focusing drop generation units are highly in demand as they overcome the major obstacle of the drop microfluidic technology. The total chip productivity of 277ghr⁻¹ is impressive. The manuscript can be accepted after minor revision.

Several comments:

Major comments:

1. The total operation time between two cleaning procedures is highly relevant for any commercial application. How long the chip can produce monodispersed droplets before it must be cleaned? Please provide the maximum operation time for both hexadecane droplets and PCL particles.
2. Biodegradable particles with 4 micron in diameter are highly in demand by pharma companies, because they can be injected directly into blood stream. Can your chip be re-designed to make 4 micron particles?

Minor comment:

-For the generation of polymer microparticles, the dispersed phase consisted of 10 wt% (133.3 g/liter) of 4 wt% (53.2 g/liter) polycaprolactone in dichloromethane. Please replace "of 4 wt%" with "or 4 wt%".

Responses to Reviewers' Comments

We thank the editor for handling the review of our paper (NCOMMS-17-25782) and thank the referees for their thoughtful and insightful feedback. Overall, the reviewers are very enthusiastic about our work. Reviewer 1 recommended that the paper be accepted after making minor revisions, Reviewer 2 noted that the “work is nicely done and solid scientifically,” and Reviewer 3 recommended that the work be accepted after making minor revisions. Reviewer 2 had some reservations about the novelty of our work relative to prior work in the field, including work done previously by our group. To address these points we have made significant additions to the text and have added significant new experimental data.

Given the enthusiasm that the reviewers had for our approach, their recognition of the significance of the problem that we have addressed, and the fact that our revisions have addressed their concerns, we hope that you will consider our work for publication. Summarized below are our responses to the reviewers' comments point-by-point. The Reviewers' comments are marked in blue, and changes to the manuscript are marked in red both here and in the manuscript.

Reviewer #1

1. This paper reports a highly parallelized microfluidic device made of silicon and glass that produces micrometer-sized drops with a narrow size distribution. Drops are produced at a throughput that is much higher than what could previously be achieved. The approximately 1000-fold increase in throughput is achieved by introducing flow resistors that make hydrodynamic resistance of the channel less dependent on its height because it mainly depends on the resistor width. The size of the resulting particles can be controlled to some degree by controlling the fluid flow rates. The main limitation of microfluidic devices that hinders their use in many scientific applications and almost all industrial applications is their low throughput. This paper presents an approach that is scalable and has thus has the potential to overcome this limitation. The paper is well written and nicely illustrated. Therefore, I recommend accepting the paper after the authors addressed the following comments.

We thank Reviewer 1 for their close read and positive assessment of our work.

2. The authors show that by changing the resistor width, they can change the maximum throughput of the device. Does the resistor width influence the drop size or drop size distribution? And how does the channel height influence the drop size?

It has been shown in previous work that the size of droplets that can be produced by a flow focusing droplet generator is comparable to that of the dimensions of the orifice (Anna et al, APL 2003). The results of our study agree with this work. The smallest droplet diameter that we could generate with our Generation I device, which had a height $h = 10 \mu\text{m}$ an orifice width $w = 10 \mu\text{m}$, was $d = 14.5 \mu\text{m}$. Whereas, the smallest droplet diameter that we could generate with our Generation III device, which had a height $h = 22.5 \mu\text{m}$ and an orifice width $w = 20 \mu\text{m}$ upstream, was $d = 22.5 \mu\text{m}$. The resistor width, which was upstream of the flow focusing droplet generator, as expected did not have an observed effect on the droplet size. To clarify this point, in the **Results** section we now write:

The size of the droplets that could be produced by each of the chips was comparable to that of the orifice dimension, as has been shown in previous work. Using our Generation III chip, which had orifice dimensions of height $h = 22.5 \mu\text{m}$ and width $w = 20 \mu\text{m}$, we were able to produce droplets with diameters $d = 22.5 \mu\text{m}$ to $37.5 \mu\text{m}$. Whereas, the Generation I chip that had orifice dimensions of height $h = 10 \mu\text{m}$ and width $w = 10 \mu\text{m}$, we were able to produce droplets with diameters $d = 14.5 \mu\text{m}$ to $18.4 \mu\text{m}$.

3. The authors use low viscosity fluids to test the performance of this device. In many cases, microparticles are made from fluids that are more viscous. Can fluids with higher viscosities also be processed and if yes, up to what viscosity? How does the maximum throughput of the device depend on the viscosity of the inner and outer phase?

Because our device is fabricated using anodically bonded Silicon and glass, it can operate at extremely high pressures (>1000 PSI), which makes it well suited for high throughput processing of highly viscous fluids. To demonstrate this functionality, we have added an experiment demonstrating the high throughput production of mineral oil droplets (30 cP), which is an order of magnitude more viscous than the hexadecane (3.0 cP) and DCM (0.41 cP) that we had previously demonstrated. Using mineral oil, we achieved a dispersed phase throughput of $\phi_D = 2.2$ L/hr ($d = 24.3$ μm droplets, $\text{CV} < 3\%$). (**FigureS7**, **ESI Video Movie_S6**) To accomplish this throughput, we used a pressure of 80 PSI on both the dispersed phase and the continuous phase, the maximum pressure that we could safely achieve with our current instrumentation (pressure vessels, tubing, flow-meters, etc...). Because the volumetric flow rate at a given pressure is inversely dependent on the viscosity, high throughput processing of highly viscous fluids requires high pressure. To achieve high flow rate processing (i.e. $\phi_D > 1.0$ L/hr) on even more viscous fluids (i.e. $\text{cP} > 100$ cP), we could replace our pressure vessels, tubing, and flow-meters to the appropriate, and available, equipment. As an example, at a pressure of 1 kPa, we project that we could process a fluid with a viscosity of 300 cP at a production rate of 2.0 L/hr.

We now clarify the utility of our device for processing highly viscous fluids in the **Results**:

Because our device is fabricated using anodically bonded Silicon and glass, it can operate at extremely high pressures (>1000 PSI), making it very well suited for high throughput processing of highly viscous fluids.^[37] We demonstrated the high throughput production of mineral oil droplets (30 cP), which is an order of magnitude more viscous than the hexadecane (3.0 cP) and DCM (0.41 cP). We achieved a dispersed phase throughput of $\phi_D = 2.2$ L/hr ($d = 24.3$ μm droplets, $\text{CV} < 3\%$). (**Fig. S7 and ESI Video 6**) The size of the droplets that could be produced by each of the chips were comparable to that of the orifice dimensions, as has been shown in previous work. Using our Device III chip, which had orifice dimensions of height $h = 22.5$ μm and width $w = 20$ μm , we were able to produce droplets with diameters $d = 22.5$ μm to 37.5 μm . Whereas, the Device I chip that had orifice dimensions of height $h = 10$ μm and width $w = 10$ μm , we were able to produce droplets with diameters $d = 14.5$ μm to 18.4 μm .

Figure S7: High throughput production of a dispersed phase of high viscosity. Dispersed phase (Φ_d) is mineral oil (30 cP) and continuous phase (Φ_c) is 2wt% Tween80 in water. The dispersed phase flow rate was $\Phi_d = 2.2$ L/hr and the continuous phase flow rate was $\Phi_c = 11.5$ L/hr. a. A micrograph of the droplets. The scale bar is 75 μm . b. A histogram of $N = 500$ droplets. The droplets had a mean diameter of 24.3 μm and a coefficient of variation $CV = 2.3\%$.

Reviewer #2:

1. This manuscript reports large-scale integration of microfluidic droplet generators. The authors lithographically microfabricated $\sim 10,000$ flow-focusing droplet generators in a matrix array on a glass/silicon chip and demonstrated mass-production of highly monodisperse emulsion droplets and particles. This work is nicely done and solid scientifically. The text is logically written having few confusing points.

We thank Reviewer 2 for their careful read and positive assessment of our work.

2. The reviewer does not consider this work represents important advances of significance necessary for publication in Nature communications. There have been several reports on similar parallelization of microfluidic droplet generators in matrix arrays from the authors' group (Lab Chip 2013, 13, 4750; Lab Chip 2015, 15, 4387) and a few other groups (Ind. Eng. Chem. Res. 2009, 48, 8881; Lab Chip 2012, 12, 802). From the reviewer's viewpoint, it seems that the design principle described in the present work is almost the same as these previous works. In particular, the authors previously reported parallelization of ~ 1000 flow-focusing droplet generators in a polymer device by soft lithography. The increase of the number up to $\sim 10,000$ based on similar approach with different device materials (glass/silicon) and conventional lithographical processes does not seem to be a significant advance.

We thank the author for their feedback; however, we respectfully disagree on their assessment of the novelty and significance of our work and would like to highlight that the other two reviewers believe that this work should be accepted with revisions. To address this concern, we now better clarify in the text the multiple fundamental innovations that were required to achieve the industrial scale production of templates for solid microscale particles (1,000X higher production rate than previously reported^{12,25}) that are appropriate for use by the pharmaceutical industry.

We have summarized below the main points:

1. To scale from prior work that achieved 1,000 flow focusing droplet generators to the 10,000 achieved in this work and beyond, we had to invent a new design principal to decouple the competing needs to keep the device resistance high for uniform flow distribution while keeping the flow focusing geometry large, so that many devices can be incorporated and each of them can operate in the dripping regime at high flow rates. The high aspect ratio resistors, fabricated by the DRIE fabrication and not easily achievable using soft lithography, enabled this decoupling by allowing us to fabricate flow resistors upstream of the microfluidic droplet generators, where the high resistance is achieved with a narrow width rather than a shallow height. The prior technologies, published by our group and others that were foundational to this work, could not achieve the throughput demonstrated here (**Fig. 1d**) that is necessary for industrial scale-up.
2. Our work presents a three-dimensional fabrication strategy that allows the scaling to 10,000 droplet generators and beyond. The prior work referenced by the reviewer, uses either two-dimensional microfluidics (Ind. Eng. Chem. Res. 2009, 48, 8881;) or two-dimensional microfluidics attached to macroscopically defined manifolds (e.g. Lab Chip 2012, 12, 802 or Lab Chip 2007, Lab Chip, 8, 287-293). These prior works laid the groundwork for ours, but our fully monolithic, three dimensionally microfabricated system is a major step forward that allows scale-up to a truly industrial scale. The monolithic construction from a single Silicon wafer obviates the alignment and bonding challenges of

prior multilayer approaches and allows high pressure use without risk of channel deformation that can plague soft elastomer devices.

3. Because our device can operate using solvents incompatible with most polymer devices, it can be used to generate high quality solid microparticles suitable for the pharmaceutical industry 1000x faster than previously achieved.^{12,25}
4. Because our device allows arbitrary microfluidic droplet generators to be parallelized, it can be designed to produce higher order emulsions and particles that require multi-step processing. To better demonstrate this modularity, we now present additional data wherein we demonstrate a VLSDI chip that incorporates 10,260 T-junction droplet generators, which can generate droplets with a CV < 4% at a throughput of 80 billion droplets per hour.(**Figure 4, ESI Video S7**)
5. Because our device can operate at high pressures and high temperatures, it can process materials (e.g. highly viscous) that could not be processed using polymer based devices. To demonstrate this functionality, we have added an experiment demonstrating the high throughput production of mineral oil droplets (30 cP), which is an order of magnitude more viscous than the hexadecane (3.0 cP) and DCM (0.41 cP) that we had previously demonstrated. Using mineral oil, we achieved a dispersed phase throughput of $\phi_D = 2.2$ L/hr ($d = 24.3$ μm droplets, CV < 3%).(**Fig S7**)

In the **Introduction**, we now clarify these points:

To address these challenges, we have developed a new approach, the all Silicon and glass Very Large Scale Droplet Integration (VLSDI) (**Figure 1a**) in which an array of 10,260 (285 x 36) microfluidic droplet generators (**Figure 1b**) can be incorporated onto a 3d-etched single silicon wafer and operated using only a single set of inlets and outlets. The monolithic construction from a single Silicon wafer obviates the alignment and bonding challenges of prior multilayer approaches and allows high pressure use.^[12-16,18] To demonstrate the power of this approach, we have generated polycaprolactone (PCL) solid microparticles, a biodegradable material approved by the US Food and Drug Administration, with a coefficient of variation CV < 5%, and an emulsion production rate that results in 277 ghr⁻¹ particle production (2.09 Lhr⁻¹ dispersed phase, 328 billion particles/hr)(**Figure 1c**), >1,000x faster than existing parallelized microfluidic approaches (**Figure 1d**).^[11,12,19] Key to achieving this throughput lies in a design strategy that breaks the trade-off between the number of integrated droplet generators N_{max} and the maximum throughput of each droplet generator ϕ'_{max} . Our design includes a high aspect ratio flow resistor into each droplet generator, which decouples the design of the individual droplet generator from the high fluidic resistance requirement necessary for parallelization.^[13-16] Moreover, because of the VLSDI's all silicon and glass construction, it can operate at high pressure ($P_{\text{max}} > 1,000$ PSI) and high temperature ($T_{\text{max}} > 500^\circ\text{C}$), use solvents prevalent in the pharmaceutical industry but that are incompatible with polymer devices, and achieve a uniformity not possible using soft-lithography based devices (CV < 3%). Because of the VLSDI's three-dimensional fabrication strategy, it can be scaled to the 10,000 droplet generators demonstrated in this study and beyond using conventional semiconductor fabrication, in contrast to prior approaches that have used either two-dimensional microfluidic^[36] or two-dimensional microfluidics attached to macroscopically defined manifolds^[12,16,32]. Because our device allows arbitrary microfluidic droplet generators to be parallelized, it can be designed to produce higher order emulsions and particles that require multi-step processing.

In the **Results**, we now highlight the modularity of our chip by demonstrating a chip that incorporates T-junction devices, in addition to the flow focusing droplet generators already presented:

Very Large Scale Integration of T-junctions onto the VLSDI chip

To demonstrate the modularity of the VLSDI architecture, we designed, fabricated, and tested a version of the chip that incorporates 10,260 (285 x 36) T-junction devices.(**Figure 4a**) This modularity allows multiple types of droplet generators, including flow focusing, T-junction, and step emulsification, as well as multiple modules to process these droplets, including pico-injectors and liquid-extraction units.^[20] We designed flow resistors (width $w = 10$ μm and height $h = 18$ μm .) upstream of the T-junctions, following the same design rules used on our FFG device.(**Figure 4b**)

We evaluated these devices by generating hexadecane droplets in water (2 wt% Tween 80) and confirmed that at all flow rates droplets are generated in every one of the 10,260 droplet generators. (ESI video S7). We found that the T-junction device could generate droplets at a maximum dispersed flow rate of $\phi_d^{\max} = 1.5$ L/hr. (Figure 4c) We mass produced oil-in-water emulsion using this device by using pressure driven flow and the size of emulsion droplets could be changed by varying the ratio of the flow rates of the dispersed and continuous phases. Droplets were generated with sizes that range from $46.2 \mu\text{m}$ ($\phi_d = 0.5$ L/hr, $\phi_c = 4.5$ L/hr) (Fig. 4d) to $33.2 \mu\text{m}$ ($\phi_d = 1.5$ L/hr, $\phi_c = 12.8$ L/hr) (Fig. 4e). The generated droplets were highly monodisperse with the coefficient of variation $CV < 4\%$ at a throughput of 80 billion droplets per hour.

We have added Figure 4:

Figure 4. A VLSI chip that contains 10,260 (285 x 36) T-junction devices. (a) A photograph of the T-junction VLSI chip. (b) An electron micrograph of four individual droplet generators. Scale bar: 250 μm . (c) Optical micrographs of T-junction droplet generators producing hexadecane droplets in water at various dispersed phase flow rates ϕ_d . The maximum flow rate ϕ_{\max} possible, while keeping the devices in the dripping regime. Scale bars of all images: 100 μm . A micrograph of hexadecane droplets in water produced by the T-junction, (d) which have an average diameter $d = 46.2 \mu\text{m}$ and a coefficient of variation $CV = 3.73 \%$ and (e) which have an average diameter $d = 33.2 \mu\text{m}$ and a coefficient of variation $CV = 3.27 \%$. Scale bar: 50 μm .

(2) The authors claim that “In contrast to prior parallelization approaches, our design breaks the trade-off... by incorporating high aspect ratio flow resistors.”. However, the insertion of flow resistors in front of each of parallel droplet generators is quite normal (e.g., see Biomicrofluidics 2013, 7, 034112; Biomicrofluidics 2015, 9, 034101).

We agree with Reviewer 2 that the use of flow resistors in microfluidics to control flow rates is well known, and has been used previously to control flow rates in droplet microfluidic systems (Biomicrofluidics 2013, 7, 034112; Biomicrofluidics 2015, 9, 034101). We now better clarify how the novelty lies in the use of flow resistors to decouple the design of individual microfluidic droplet generators for massive parallelization that requires uniform distribution of fluids across 10,200 microfluidic droplet generators, which to our

knowledge has not been previously reported. Importantly, in the two papers noted by the reviewer- flow rates were limited to <30 mL/hr and the number of FFGs to <10, and the resistors were used to control the flow rate and to minimize device coupling, which are important foundational papers but do not tackle the same challenge of massive parallelization that our paper focuses on. Moreover, we show in our own data that by using similar resistors to those used in the prior approaches noted by the reviewer (Dev. I VLSDI), we can only achieve throughputs of 0.35 L/hr with 10,260 droplet generators, compared to Dev. III that uses our decoupling strategy and can achieve 7.2 L/hr dispersed phase throughput.. In the **Results** section we now write:

This work, wherein we have, for the first time to our knowledge, decoupled the design of individual flow focusing droplet generators in massively parallelized chips, builds on earlier reports where flow resistors were used to control the flow rate and decouple droplet generators in chips that contain several droplet generators.^[30,31]

(3) The use of non-polymer materials like glass and silicon for fabricating parallel microfluidic devices for generating droplets and particles is not new (e.g., *Ind. Eng. Chem. Res.* 2002, 41, 4043; *Lab Chip* 2008, 8, 287; *Lab Chip* 2012, 12, 3426; *Lab Chip* 2015, 15, 2486; *Macromol. Chem. Phys.* 2017, 218, 1600472, etc.) . It is not surprising to use non-polymer materials together with conventional microfabrication processes although their cost is higher than those of polymer devices.

We now better clarify that the use of Silicon and glass for droplet microfluidics presented in our paper is not new. As the reviewer has noted, the use of these fabrication techniques date back to the origins of microfluidics as a field. We have clarified our claims to make it more clear that the novelty of our VLSDI device is in its design, which was enabled by the use of the properties of Si and glass (high pressure, solvent resistance) its Deep Reactive Ion Etch (DRIE) fabrication - i.e. high aspect ratio flow resistors, low variation on channel geometries for very high particle mono-dispersity, and use at high pressures to process viscous fluids (**Fig. S7**).

To address these challenges, we have developed a new approach, the all Silicon and glass Very Large Scale Droplet Integration (VLSDI) (**Figure 1a**) in which an array of 10,260 (285 x 36) microfluidic droplet generators (**Figure 1b**) can be incorporated onto a 3d-etched single silicon wafer and operated using only a single set of inlets and outlets. The monolithic construction from a single Silicon wafer obviates the alignment and bonding challenges of prior multilayer approaches and allows high pressure use.^[12-16,18] To demonstrate the power of this approach, we have generated polycaprolactone (PCL) solid microparticles, a biodegradable material approved by the US Food and Drug Administration, with a coefficient of variation $CV < 5\%$, and an emulsion production rate that results in 277 ghr⁻¹ particle production (2.09 Lhr⁻¹ dispersed phase, 328 billion particles/hr)(**Figure 1c**), >1,000x faster than existing parallelized microfluidic approaches (**Figure 1d**).^[11,12,19] Key to achieving this throughput lies in a design strategy that breaks the trade-off between the number of integrated droplet generators N_{max} and the maximum throughput of each droplet generator ϕ'_{max} . Our design includes a high aspect ratio flow resistor into each droplet generator, which decouples the design of the individual droplet generator from the high fluidic resistance requirement necessary for parallelization.^[13-16] Moreover, because of the VLSDI's all silicon and glass construction, it can operate at high pressure ($P_{max} > 1,000$ PSI) and high temperature ($T_{max} > 500^{\circ}\text{C}$), use solvents prevalent in the pharmaceutical industry but that are incompatible with polymer devices, and achieve a uniformity not possible using soft-lithography based devices ($CV < 3\%$). Because of the VLSDI's three-dimensional fabrication strategy, it can be scaled to the 10,000 droplet generators demonstrated in this study and beyond using conventional semiconductor fabrication, in contrast to prior approaches that have used either two-dimensional microfluidic^[36] or two-dimensional microfluidics attached to macroscopically defined manifolds^[12,16,32]. Because our device allows arbitrary microfluidic droplet generators to be parallelized, it can produce higher order emulsions and

particles that require multi-step processing, which could not be fabricated using competing technology (e.g. step emulsification).[18,33-36]

Additionally, we have added the following references:

- [32] T. Nisisako, T. Ando, and T. Hatsuzawa, *Lab on a Chip*, **2012**, *12*, 3426-3435.
- [33] S. Sahin and S. Karin, *Lab on a Chip*, **2015**, *15*.11, 2486-2495.
- [34] A. Ofner, et al. *Macromolecular Chemistry and Physics*, **2017**, *218*.2, 1600472.
- [35] S. Sugiura, M. Nakajima, and M. Seki. *Industrial & engineering chemistry research*, **2002**, *41*.16, 4043-4047.
- [36] G. Tetradis-Meris, et al. *Industrial & Engineering Chemistry Research*, **2009**, *48*.19, 8881-8889.

Moreover, we would like to make a note to the reviewer that Reference 18,33-36 perform step-emulsification, which unlike our design that allows arbitrary microfluidic droplet generators to be parallelized, these devices cannot produce higher order emulsions and particles that require multi-step processing. Reference 12,32 perform parallelization using a 1D array (annular) of flow focusing droplet generators that use a macroscopic fitting to deliver fluid to the individual droplet generators. Because of the VLSDI's three-dimensional fabrication strategy, it can be scaled to the 10,000 droplet generators demonstrated in this study and beyond using conventional semiconductor fabrication, in contrast to many prior approaches that have used either two-dimensional microfluidic^[36] or two-dimensional microfluidics attached to macroscopically defined manifolds^[12,16,32].

Reviewer #3

1. The manuscript describes the performance of a high-throughput microfluidic drop generation system. Microfluidic systems with more than 10,000 flow focusing drop generation units are highly in demand as they overcome the major obstacle of the drop microfluidic technology. The total chip productivity of 277ghr⁻¹ is impressive. The manuscript can be accepted after minor revision. Several comments:

We thank the reviewer for their close read and positive assessment of our work.

2. The total operation time between two cleaning procedures is highly relevant for any commercial application. How long the chip can produce monodispersed droplets before it must be cleaned? Please provide the maximum operation time for both hexadecane droplets and PCL particles.

We have produced as much as 2 gallons of hexadecane/water emulsion routinely, at a dispersed flow rate of $\phi_d = 7.3 \text{ Lhr}^{-1}$ and a run time of half of an hour without a single device failing. And, we have routinely produced as much as two gallons of DCM-PCL/water emulsion at a flow rate of $\phi_d = 2.09 \text{ Lhr}^{-1}$ and a run time of seventy five minutes without a single device failing. These tests were limited by the size of our pressure vessels (1 gallon dispersed, 3 gallons for continuous phase). We posit that because we use high quality reagents, in-line filters upstream of the device input (**Fig. S5**), and steel compression fittings rather than conventionally used syringe-tip filters (**Fig. S4**), there were no particulates to clog the device. In an industrial setting, similar instrumentation could be used and in-line quality control can be implemented to stop the device if a blockage occurs. We have now added to the **Results** section the following:

We have produced as much as two gallons of DCM-PCL/water emulsion at a flow rate of $\phi_d = 2.09 \text{ Lhr}^{-1}$ with a run time of seventy five minutes without a single device failing. These tests were limited by the size of our pressure vessels -1 gallon dispersed, 3 gallons for continuous phase (**Fig. S5**).

3. Biodegradable particles with 4 micron in diameter are highly in demand by pharma companies, because they can be injected directly into blood stream. Can your chip be re-designed to make 4 micron particles?

Yes, we thank the reviewer for this valuable suggestion and we will pursue it in future work. We believe that we can make these particles using our current chips, by both reducing the droplet diameters that we use and/or the concentration of PCL, as we show in our calibration curve in **Fig. 5I**.

4. For the generation of polymer microparticles, the dispersed phase consisted of 10 wt% (133.3 g/liter) of 4 wt% (53.2 g/liter) polycaprolactone in dichloromethane. Please replace "of 4 wt%" with "or 4 wt%".

We thank the reviewer for their close read. We have updated the text.

Reviewers' Comments:

Reviewer #1:

Remarks to the Author:

The authors added experimental data to clarify their findings and made modifications to their manuscript that fully answer the concerns I had. There is just one minor thing that I would like to ask the authors to change: In the change they made in response to the third question of referee 1, the authors state that "We demonstrated the high throughput production of mineral oil droplets (30 cP), which is an order of magnitude more viscous than the hexadecane (3.0 cP) and DCM (0.41 cP)." The viscosity of DCM is actually two orders of magnitude higher than that of mineral oil. The authors should clarify this for consistency. After the authors made this minor change, I recommend accepting this paper for publication in Nature communications.

Reviewer #3:

Remarks to the Author:

The revised manuscript can be accepted in its current form.